# Histidine-Rich Glycoprotein Suppresses the S100A8/A9-Mediated Organotropic Metastasis of Melanoma Cells

**DOI:** 10.3390/ijms231810300

**Published:** 2022-09-07

**Authors:** Nahoko Tomonobu, Rie Kinoshita, Hidenori Wake, Yusuke Inoue, I Made Winarsa Ruma, Ken Suzawa, Yuma Gohara, Ni Luh Gede Yoni Komalasari, Fan Jiang, Hitoshi Murata, Ken-ichi Yamamoto, I Wayan Sumardika, Youyi Chen, Junichiro Futami, Akira Yamauchi, Futoshi Kuribayashi, Eisaku Kondo, Shinichi Toyooka, Masahiro Nishibori, Masakiyo Sakaguchi

**Affiliations:** 1Department of Cell Biology, Okayama University Graduate School of Medicine, Dentistry and Pharmaceutical Sciences, Okayama 700-8558, Japan; 2Department of Pharmacology, Kindai University Faculty of Medicine, Osaka 589-0014, Japan; 3Faculty of Science and Technology, Division of Molecular Science, Gunma University, Kiryu 376-8515, Japan; 4Faculty of Medicine, Udayana University, Denpasar 80232, Indonesia; 5Department of General Thoracic Surgery and Breast and Endocrinological Surgery, Okayama University Graduate School of Medicine, Dentistry and Pharmaceutical Sciences, Okayama 700-8558, Japan; 6Department of General Surgery & Bio-Bank of General Surgery, The Fourth Affiliated Hospital of Harbin Medical University, Harbin 150001, China; 7Department of Interdisciplinary Science and Engineering in Health Systems, Okayama University, Okayama 700-8530, Japan; 8Department of Biochemistry, Kawasaki Medical School, Kurashiki 701-0192, Japan; 9Division of Molecular and Cellular Pathology, Niigata University Graduate School of Medical and Dental Sciences, Niigata 951-8510, Japan; 10Division of Tumor Pathology, Near InfraRed Photo-ImmunoTherapy Research Institute, Kansai Medical University, Osaka 573-1010, Japan; 11Department of Translational Research & Drug Development, Okayama University Graduate School of Medicine, Dentistry and Pharmaceutical Sciences, Okayama 700-8558, Japan

**Keywords:** S100A8/A9, HRG, metastasis

## Abstract

The dissection of the complex multistep process of metastasis exposes vulnerabilities that could be exploited to prevent metastasis. To search for possible factors that favor metastatic outgrowth, we have been focusing on secretory S100A8/A9. A heterodimer complex of the S100A8 and S100A9 proteins, S100A8/A9 functions as a strong chemoattractant, growth factor, and immune suppressor, both promoting the cancer milieu at the cancer-onset site and cultivating remote, premetastatic cancer sites. We previously reported that melanoma cells show lung-tropic metastasis owing to the abundant expression of S100A8/A9 in the lung. In the present study, we addressed the question of why melanoma cells are not metastasized into the brain at significant levels in mice despite the marked induction of S100A8/A9 in the brain. We discovered the presence of plasma histidine-rich glycoprotein (HRG), a brain-metastasis suppression factor against S100A8/A9. Using S100A8/A9 as an affinity ligand, we searched for and purified the binding plasma proteins of S100A8/A9 and identified HRG as the major protein on mass spectrometric analysis. HRG prevents the binding of S100A8/A9 to the B16-BL6 melanoma cell surface via the formation of the S100A8/A9 complex. HRG also inhibited the S100A8/A9-induced migration and invasion of A375 melanoma cells. When we knocked down HRG in mice bearing skin melanoma, metastasis to both the brain and lungs was significantly enhanced. The clinical examination of plasma S100A8/A9 and HRG levels showed that lung cancer patients with brain metastasis had higher S100A8/A9 and lower HRG levels than nonmetastatic patients. These results suggest that the plasma protein HRG strongly protects the brain and lungs from the threat of melanoma metastasis.

## 1. Introduction

Cancer metastasis is a crucial and imperative subject because of its threat to human life. It is thus very important to clarify the metastatic processes at the cellular and molecular levels in order to establish cancer management strategies that effectively prevent metastasis. However, the full elucidation of these processes remains elusive, largely because they are so complex, involving interplay between cancer cells and various elements of the surrounding cancer milieu including inflammatory immune cells, fibroblasts, endothelial cells, and extracellular matrices. These multilayered interactions greatly contribute to the spatiotemporal patterns of the metastatic steps, which are mediated by the orchestrated cross-talk among numerous extracellular and intracellular molecules. Among the veritable mountain of molecules associated with cancer metastasis, S100A8/A9, a heterodimer complex of S100A8 and S100A9 proteins, is especially noteworthy. The functional blocking of extracellular S100A8/A9 by the anti-S100A8/A9 neutralizing antibody Ab45, which we developed, efficiently prevents lung-directed melanoma metastasis even after a single injection [1]. Moreover, we have reported an unusual role of S100A8/A9 in the metastasis of lung-tropic cancer—namely, binding to the cell surface S100 soil sensor receptors (SSSRs). The SSSR family includes such classical receptors as toll-like receptor 4 (TLR4) [2,3] and the receptor for advanced glycosylation end products (RAGE) [4,5,6], as well as novel receptors such as extracellular matrix metalloproteinase inducer (EMMPRIN) [7,8], neuroplastin (NPTN) α and β (NPTNβ compensates for NPTNα) [8], activated leukocyte cell adhesion molecule (ALCAM) [8], and melanoma cell adhesion molecule (MCAM) [9]. These individual receptors show specific expression patterns according to the cancer cell type and grade [10]. For example, TLR4, RAGE, and EMMPRIN were reported to be expressed in metastatic melanoma cells [9,11,12], and the EMMPRIN-mediated stimulation of melanoma cells with the extracellular S100A8/A9 secreted from the lung strongly facilitated the remote metastasis of melanoma cells to the lung [7]. Additionally, MCAM is abundant in metastatic melanoma and breast cancer cells of the worst grade; in these cells, MCAM readily recognizes and binds to extracellular S100A8/A9, which induces lung metastasis through the activation of the MCAM downstream pathways—i.e., the tumor progression locus 2 (TPL2)-ETS variant transcription factor 4 (ETV4)–matrix metallopeptidase 25 (MMP25) axis in the case of melanoma and the (TPL2)–(ETV4)–zinc finger E-box binding homeobox 1 (ZEB1) axis for breast cancer, both of which provide cancer cells with a metastatic driving force [13,14]. NPTNβ is highly expressed in lung cancers. The stimulation of NPTNβ on lung cancer cells with S100A8/A9 leads to an invasive event in the lung via the activation cascade of nuclear factor I (NFI)A/NFIB and SAM pointed domain-containing ETS transcription factor (SPDEF) with the help of two adaptor proteins: tumor necrosis factor (TNF) receptor-associated factor 2 (TRAF2) and growth factor receptor-bound protein 2 (GRB2) [15]. Thus, an elevated level of S100A8/A9 is a critical danger signal triggering cancer metastasis via certain molecular mechanisms. 

S100A8/A9 is highly expressed in neutrophils as well as in monocytes. It has been reported that more than 40% of the cytosolic proteins in neutrophils are S100A8/A9 [16]. The expression of S100A8/A9 is strongly induced by tissue injuries [17,18]. The massive release of S100A8/A9 from these sources will induce the elevation of plasma levels of S100A8/A9, resulting in the initiation of inflammatory responses as a damage-associated molecular pattern (DAMP) and the modulation of cancer metastasis as mentioned above. However, little is known about the existence of anti-DAMP molecules for S100A8/A9. There are no reports at present about any physiological molecules that bind to the extracellular S100A8/A9 and counteract the S100A8/A9-triggered metastatic activity. However, we previously showed that a neutralizing anti-S100A8/A9 antibody conferred protection against both cancer metastasis and inflammatory responses [1]. Such an endogenous molecule may play protective and homeostatic roles in the extracellular space. To address this possibility, in the present study, we performed an affinity purification of candidate molecules that bind to S100A8/A9 and antagonize the effects of S100A8/A9. We found that a plasma glycoprotein, histidine-rich glycoprotein (HRG), is a molecule that efficiently regulates the activity of extracellular S100A8/A9.

## 2. Results

### 2.1. Identification of S100A8/A9 Binding Protein(s) in Human Plasma

To “fish” for S100A8/A9 binding proteins, we prepared GST-fused S100A8 and S100A9 recombinant proteins as bait. We applied these proteins to a pull-down purification assay using glutathione-conjugated beads and healthy human plasma as a source of interacting proteins. As shown in Figure 1A, we succeeded in detecting three distinct candidate bands on Coomassie brilliant blue staining of SDS-PAGE gels, BP1, BP2, and BP3, as indicated by the filled circles in Figure 1A, that co-precipitated with the combination of GST-S100A8 and GST-S100A9 or with GST-S100A9 alone but not with GST-S100A8, indicating that the proteins corresponding to these three bands bind with the S100A8/A9 heterodimer complex at the S100A9 side. Except for a 40-kDa band, we detected no nonspecific band from human plasma under the conditions used in these experiments. Mass spectrometric analysis and correspondence with data in the SWISS-PROT database unveiled bound proteins. Interestingly, they were all identified as the same protein, HRG (Appendix A). Then, the lower two bands (BP2 and BP3) were estimated to be proteolytic products of BP1. The plasma protein HRG is composed of 525 amino acids before processing and secretion with a total of about 20-KDa glycosyl chains. HRG has a highly prominent domain: a central histidine-rich region flanked by two proline-rich regions. This histidine-rich domain may allow HRG to interact with many molecules [19,20].

To confirm this novel interaction between the S100A8/A9 heterodimer and plasma glycoprotein HRG, the binding was further studied with enzyme-linked immunosorbent assay (ELISA) (Figure 1B) and immunoprecipitation (IP) (Figure 1C) analysis. The recombinant human S100A8/A9 heterodimer complex was found to bind to the HRG protein purified from human plasma and immobilized on the ELISA plate in a concentration-dependent manner (Figure 1B). For additional binding analysis, we employed an IP method developed in our lab using the membrane-anchored HRG and S100 proteins (Figure 1C, left image) [21]. This is an excellent method for studying the interactions between extracellular secretory proteins and secretory proteins, as well as those between extracellular secretory proteins and membrane proteins, because the transfected genes are expressed at high density on the cell surface as tethered ligands without diffusing into the vast extracellular fluid space. Using this approach (the pull-down result shown in Figure 1A), HRG was able to bind to S100A9 with the highest affinity (Figure 1C, right). We also found that other S100 family proteins, specifically S100A3, S100A4, and S100A5, bound to HRG. The longer exposure time (Figure 1C, bottom) on Western blotting enabled us to detect the S100 proteins with lower affinities for HRG. There were no signals on GFP, a negative control, or on the remaining S100 proteins; therefore, the positive signals represented the specific reactions due to interaction between S100 proteins and HRG. Taken together, these results suggest that HRG can bind to S100 family proteins with different affinities.

### 2.2. Inhibitory Effects of HRG on S100A8/A9-Mediated Melanoma Cell Behaviors In Vitro

Next, to evaluate the significance of HRG binding to S100A8/A9, we used cancer cells in culture with the purified S100A8/A9 recombinant protein in the presence or absence of HRG in the extracellular space. The results showed that HRG quenches the S100A8/A9 binding to the surfaces of both B16-BL6 mouse melanoma cells (Figure 2A) and A375 human melanoma cells (Figure 2B), suggesting that S100A8/A9 loses its binding force to cell surface SSSRs via the formation of a complex with HRG. Consistent with the prevention of the surface binding of S100A8/A9 by HRG, the addition of HRG to medium diminished the S100A8/A9-mediated migration and invasion in A375 cells (Figure 2C). These results suggest that HRG has an antagonistic effect on extracellular S100A8/A9 through complex formation (if not through other as-yet unknown mechanisms as well) that will deprive S100A8/A9 of its ability to bind to its receptor(s) on the cells.

### 2.3. Preventive Roles of HRG on Cancer Growth and Metastasis In Vivo

To evaluate the effects of plasma HRG on cancer metastasis in vivo, we knocked down the synthesis of HRG in the liver by Hrg siRNA. A syngeneic and orthotopic allograft model of melanoma was employed to elicit an autonomous metastasis and to evaluate the effect of HRG on that metastasis. According to the established procedure, we first inoculated melanoma cells into a subcutaneous area of the mouse ear on one side and then on the 18th day post-inoculation further treated the mice with the validated mouse Hrg siRNA (siHrg) to reduce the plasma HRG protein [22]. We previously reported that this siHrg method induced a long-lasting reduction in plasma HRG levels of at least 6 days after injection [22]. One day after siHrg treatment, we artificially injured the growing tumors by excising a portion of the tumor equivalent to one half the size necessary to facilitate metastatic onset and growth acceleration (Figure 3A). On day 13, the final day of the in vivo study schedule, we confirmed that the plasma levels of HRG in the mice were all significantly downregulated by the transduction of siHrg but not by that of the control siRNA (siCont) (Figure 3B). The reduction in plasma HRG was associated with an accelerated outgrowth of melanoma (Figure 3C). Moreover, the fierce growth of melanoma in the siHrg-treated mice induced not only lung metastasis but also brain metastasis at significant levels (Figure 3D, left (images) and right (quantified data)).

### 2.4. Significant Increase in S100A8/A9 at Premetastatic Stage in the Lungs and Brain of Melanoma-Bearing Mice

Figure 4A shows the protocol of qPCR for S100 family members in the lungs and the brain in the premetastatic phase (on day 7 after inoculation of melanoma in the ear skin). S100A8/A9 expression was remarkably and specifically increased in S100 family members in both the lungs and brain before the onset of metastatic outgrowth in those organs at a relatively early time point (day 7) after the intradermal injection of melanoma cells (Figure 4B).

Finally, to consolidate these in vitro and in vivo findings, we investigated the plasma levels of S100A8/A9 and HRG in clinical patients who suffered from brain metastasis of lung cancer. It should be noted that the plasma levels of S100A8/A9 and HRG were in a reciprocal relationship—that is, S100A8/A9 was significantly elevated while HRG was significantly reduced in the patients with brain metastasis compared with the levels in healthy donors (Figure 4C).

## 3. Discussion

In this study, we identified HRG as a novel plasma protein that binds directly to and inhibits the heterodimer S100A8/A9, effectively preventing S100A8/A9-mediated organotropic cancer metastasis [1]. HRG is present at relatively high concentrations (60–100 µg/mL) in human blood [23], and the protein is produced mainly in the liver [24]. HRG plays a crucial role in multiple important biological processes, including angiogenesis [21,25,26], coagulation [27], and pathogen/dead cell clearance [28,29,30], through its interactions with multiple molecules under physiological and pathological settings in the human body [20]. Surprisingly, however, there have been no reports on HRG’s interaction with S100A8/A9 or on the relevance of HRG to S100A8/A9-mediated cancer biology. The local and systemic balance between HRG and S100A8/A9 may regulate the migration, invasion, and metastasis of melanoma and could be a determinant of cancer pathogenesis leading to an aggressive phenotype.

In discussing our present findings, the first result to consider is the appearance of HRG bands with three different molecular sizes in our pull-down purification procedure using GST-fusion proteins and healthy human plasma as the starting material (Figure 1A). The diversity of molecular sizes was probably due to the unintended proteolytic cleavage of HRG occurring through the procedure itself, since the protein is readily subjected to proteolysis in the presence of heparin [19], which was added to the blood specimens prior to plasma preparation. In addition, the reaction mixture contained no additional low-molecular-weight protease inhibitors from commercial sources.

The second point to consider is that HRG was downregulated in our patients with brain-metastatic lung cancer (Figure 4C), which would lead to the circumstance that S100A8/A9 was dominant in the cancer milieu at the onset site as well as in the cancer premetastatic environments in the remote organs. Previously we reported that septic inflammation leads to the downregulation of HRG at not only the mRNA level in the liver but also the protein level in the circulating blood by the deposition on thrombi and increased proteolytic degradation [22]. Septic inflammation may induce a signal that suppresses HRG transcription. In fact, we also found a significant reduction of mouse Hrg mRNA in liver resected from melanoma-bearing mice even early on the 7th day after inoculation of melanoma cells (Appendix A). There is little information about the molecular mechanism underlying the transcriptional regulation of HRG in hepatocytes in the liver; therefore, our investigation into the downregulation of HRG under cancer-mediated inflammation is ongoing. On the other hand, the HRG protein level in the blood is known to be regulated by proteolytic degradation by inflammation-mediated thrombin activation under septic conditions [22]. We therefore consider that a mechanism of HRG degradation similar to that in a septic setting could occur in the context of cancer-mediated inflammation, which would further increase in the malignant metastatic stages since metastatic invasion is severely injurious to normal tissues.

Another aspect that deserves discussion is the increase in S100A8/A9 levels in the premetastatic distant organs, i.e., the lungs and brain, in the nonmetastatic early stage in our experiments (Figure 4B). Hoshino et al. reported that various S100 genes (S100A4, A6, A10, A11, A13, and A16) are induced simultaneously in normal fibroblasts in the lungs in response to stimulation with tumor-derived exosomes [31]. In light of a unique trait shared by all secretory S100 proteins—namely, that they induce inflammatory cytokines and chemokines—cancer-induced S100 proteins produced in lung fibroblasts would add an additional source of S100 proteins to augment the main sources of S100A8/A9 secretory production: neutrophils, monocytes, and macrophages. Taken together, these results show that the induction and increased secretion of S100A proteins function mutually to foster an inflammatory environment in the lungs. The induction of S100A8/A9 in the premetastatic phase in the brain was remarkable, like that in the lung. It has been reported that microglia and astrocytes are potential sources of S100A8/A9 secretory production in the brain of cancer-bearing animals [32,33]. It might be possible that cancer-derived exosomes provoke brain inflammation via high levels of S100A8/A9 in these specific cells after passing through the blood-brain barrier (BBB). However, the exact mechanism underlying S100A8/A9 induction has still not been fully elucidated, so this is also a matter of ongoing investigation in our laboratory.

The notable anti-cancer effect of HRG in vivo leads to our last point of discussion. Given the multiple molecules that interact with HRG, HRG-mediated cancer prevention in vivo would not be attributable to the prevention of S100A8/A9 activities alone. As mentioned in the Introduction, the S100 soil sensor receptors (SSSRs) in cancer cells include the classical receptors TLR4 and RAGE [11]. These receptors could be activated by another representative DAMP ligand, high mobility group box 1 (HMGB1), leading to metastasis in several sorts of cancer cells. Interestingly, in endothelial cells, HRG can bind to HMGB1 [34], inhibit HMGB1-mediated angiogenesis [22], and antagonize HMGB1-mediated cellular events [21]. In addition, we found that HRG may bind with S100 proteins other than S100A8/A9. One of these proteins, extracellular S100A4, has been shown to play unusual roles in cancer progression [35] that are very similar to the roles of HMGB1 and S100A8/A9. S100A4 secreted in cancer tissues may play roles in the acceleration of inflammation, angiogenesis, invasive motility, and fibrosis. Collectively, those interactions could play significant roles in HRG-mediated anti-cancer effects. We also should not disregard another function of HRG—namely, as a ligand for certain cell surface receptors. We have reported that HRG stimulates cell surface C-type lectin family 1A (CLEC1A), which is expressed in vascular endothelial cells and neutrophils [21]. HRG can prevent HMGB1 secretion from endothelial cells by binding to CLEC1A, leading to the diminution of cytokine responses in endothelial cells [21]. Via CLEC1A, HRG also stimulates neutrophils to superior performance as seen in cancer-inhibiting N1-neutrophils, which exhibit prolonged longevity [36], implying that HRG may sustain neutrophils in an N1 state. On the other hand, another type of neutrophils called N2 neutrophils appear to be tightly involved in cancer progression through the active formation of neutrophil extracellular traps (NETs) [37], which release abundant HMGB1 extracellularly as well as S100A8/A9 [38,39] and our unpublished data. Thus, it seems reasonable to speculate that HRG may also function to block cancer activation by regulating young neutrophils. In the cancer milieu, macrophages, especially M2-type macrophages, play a crucial role in fostering cancer cells toward their aggressive metastatic forms [40]. Rolny et al. reported that HRG can regulate the polarization of tumor-associated macrophages (TAM) from M2-like type to cancer-inhibitable M1-like type [41]. In addition, HRG activates natural killer (NK) cell–cancer cell cytotoxicity through the downregulation of programmed-death-1 (PD-1), a cancer immune tolerance mediator, on the NK cell surface. This effect may be mediated by HRG-CLEC1B (another receptor of HRG we identified) binding [42]. Taken together, these facts suggest that multifunctional HRG may not only control cancer cells but also regulate the surrounding milieu, where endothelial cells, neutrophils, macrophages, and NK cells are closely involved in cancer metastatic progression. The ability of HRG to capture important molecules, including S100A8/A9, HMGB1, and probably other molecules such as S100A4, should play an important role in the regulation of cancer progression and metastasis, in addition to eliciting effects through the stimulation of specific receptors such as CLECs. We have already begun to study these important subjects.

In conclusion, our results indicate that HRG plays crucial roles in inhibiting cancer growth and metastasis through its absorptive binding with extracellular S100A8/A9. Considering the clinical data showing a reciprocal relationship between HRG and S100A8/A9, their balance in cancer patients is one of the few potent determinants of a shift in cancer metastasis to a highly aggressive state. Based on how effective HRG is against S100A8/A9, HRG may lead to an effective therapy to prevent the progression of several types of cancers.

## 4. Materials and Methods

### 4.1. Cell Lines

B16-BL6 cells (a highly invasive variant of the mouse malignant melanoma B16 cell line; a kind gift from Dr. Isaiah J. Fidler of the MD Anderson Cancer Center, Houston, TX, USA) were cultured in D/F medium (Thermo Fisher Scientific, Waltham, MA, USA) supplemented with 10% FBS in a humidified incubator. A375 cells (a human malignant melanoma cell line; ATCC, Rockville, MD, USA) were also used in this study. A375 cells were cultured with RPMI medium (Thermo Fisher Scientific) supplemented with 10% FBS. All cultures were checked for mycoplasma by using both a mycoplasma detection kit (Thermo Fisher Scientific) and Hoechst 33,342 staining at regular intervals.

### 4.2. Liquid Chromatography-Mass Spectrometry (LC-Ms/Ms)

The CBB-stained bands, BP1, 2, and 3, were cut out from the gel, and their individual pieces were individually trypsinized in a trypsin digestion buffer (10 mM CaCl2, 100 mM ammonium bicarbonate, pH 7.8) overnight at 37 °C. The digested liquid specimens were then subjected to a protein identification procedure using a nano-flow liquid chromatography–mass spectrometry apparatus (Agilent 6330 Ion Trap; Agilent Technologies, Santa Clara, CA, USA) equipped with an analytical chip (Agilent HPLC-Chip; Agilent Technologies). The resulting tandem mass spectrometry spectra of the tryptic peptides were analyzed using Agilent software (Spectrum Mill MS Proteomics Workbench; Agilent Technologies) with a protein database (SWISS-PROT).

### 4.3. Enzyme-Linked Immunosorbent Assay (ELISA)

To evaluate the binding between S100A8/A9 and HRG, an ELISA-based assay was employed. The purified human plasma HRG (100 μg/mL) was homogeneously plated on the bottoms of the wells of a 96-well plate. After washing the wells with T-PBS (0.05% Tween 20 in PBS) and subsequent blocking with Blockmaster DB1130 (MBL, Nagoya, Japan), different concentrations of biotinylated human recombinant S100A8/A9 protein were actuated with the HRG protein attached to the bottoms of the wells. The bound S100A8/A9 was detected by treatment with HRP-conjugated streptavidin and the subsequent reaction with a substrate, 3, 3′, 5, 5′-tetramethylbenzidine (TMB) (1-Step™ Ultra TMB-ELISA Substrate Solution (Thermo Fisher Scientific)).

Next, the levels of S100A8/A9 and HRG in human blood specimens were evaluated. Plasma samples were collected from January 2012 through March 2020 at Okayama University Hospital from patients with lung cancers that had metastasized to the brain. The characteristics of the patients are summarized in Table 1. Human S100A8/A9 and HRG ELISA were performed using a sandwich procedure under conventional conditions using a mouse anti-human S100A8/A9 antibody (clones #45 and #260) developed by our group, as reported previously, and a commercially available HRG-ELISA kit (human HPRG/HRG ELISA pair set; Sino Biological, Beijing, China).

### 4.4. Recombinant Proteins

The highly purified human recombinant S100A8/A9 heterodimer was prepared as described previously [43]. The glutathione transferase (GST)-fusion human S100A8 and S100A9 proteins were also prepared as previously reported [44]. The human plasma HRG protein was isolated at high purity by the established method using Ni-NTA affinity (Thermo Fisher Scientific) and Mono-Q columns from whole blood specimens collected by healthy volunteer donors as previously reported [22].

### 4.5. Plasmids

All mammalian gene expression plasmids used were constructed using the pIDT-SMART (C-TSC) (pCMViR-TSC) vector as the backbone to express the cargo genes at high levels. All the plasmids were designed to be expressed as a cell surface-presented form with C-terminal 3myc-6His-tag or 3HA-6His-tag [45]. Transient transfection of these plasmids into cultured cells was performed using FuGENE-HD (Promega BioSciences, San Luis Obispo, CA, USA).

### 4.6. Immunoprecipitation (IP)

Monoclonal Anti-myc tag agarose (MBL) was used for the immunoprecipitation experiments to capture the ectopically overexpressed proteins. The tag-agarose beads were mixed with plasma and incubated for 3 h at 4 °C. After the incubation of the individual samples, bound proteins were pulled down by centrifugation. The precipitated proteins were subjected to SDS-PAGE and detected with Western blotting with mouse anti-myc tag antibody (Cell Signaling Technology, Danvers, MA, USA) or anti-HA tag antibody (Cell Signaling Technology).

### 4.7. Western Blotting (WB)

WB analysis was performed under conventional conditions. The antibodies used were as follows: rabbit anti-human S100A8 (Calgranulin A) polyclonal antibody (#sc-20174; Santa Cruz Biotechnology, Dallas, TX, USA), rabbit anti-human S100A9 (Calgranulin B) polyclonal antibody (#sc-20173; Santa Cruz Biotechnology), rabbit anti-human HRG antibody (#PA5-97051; Thermo Fisher Scientific) that cross-reacts with mouse HRG, and mouse anti-tubulin monoclonal antibody (#T5168, clone B-5-1-2; Sigma-Aldrich, St. Louis, MO, USA).

### 4.8. Migration and Invasion Assays

Cell invasion or migration was assayed using the Boyden chamber method with filter inserts (pore size, 8 μm) pre-coated or not pre-coated with Matrigel in 24-well plates (BD Biosciences, Franklin Lakes, NJ, USA). A375 cells (5 × 10^4^ cells/insert) were seeded with a low-serum RPMI medium (0.5% FBS) on the top chamber, and the bottom chamber was filled with a 10% FBS-supplemented conventional RPMI medium. Recombinant S100A8/A9 was then set in the bottom chamber at a final concentration of 1 μg/mL with purified HRG protein or BSA at a final concentration of 100 μg/mL. In melanoma patients, although the level of S100A8/A9 in serum was previously reported to be 1–15 μg/mL [34], our study showed that 1 μg/mL of S100A8/A9 had the highest levels of migration and invasion activity in A375 cells in the Boyden chamber method (our unpublished data). We therefore decided to use S100A8/A9 at 1 μg/mL. On the other hand, because of the relatively high concentration of HRG in healthy human serum (60–100 μg/mL) [23], we used HRG at a final concentration of 100 μg/mL in this evaluation. After incubation for 18 h, cells that passed through the filter were counted by staining with hematoxylin and eosin (H&E). Migrated/invaded cells were imaged under a microscope (BZ-9000; Keyence, Tokyo, Japan), quantified by cell counting in five non-overlapping fields at ×100 magnification, and presented as the average from three independent experiments.

### 4.9. Quantitative Real-Time PCR

Total RNA was extracted from the resected mouse organs using ISOGEN II Isolation Reagent (Nippon Gene, Tokyo, Japan). Reverse transcription was then performed using ReverTraAce qPCR RT Master Mix with gDNA Remover (Toyobo, Osaka, Japan). A real-time polymerase chain reaction (PCR) was performed using FastStart SYBR^®^ Green Master Mix (Roche Applied Science, Penzberg, Germany) with specific primers on a StepOnePlus™ Realtime PCR system (Applied Biosystems, Foster City, CA, USA). The primers used are shown in Table 2.

### 4.10. Animal Experiment

To establish lung-tropic cancer metastasis in an orthotopic autonomous manner, B16-BL6 cells (1 × 10^5^ cells) were intradermally injected into the back side of one ear (day -18). After the tumors had grown to 4–5 mm in diameter for 18 days, the validated siHrg (sense: GUUCUAGACCUGAUCAAUAtt; antisense: UAUUGAUCAGGUCUAGAACtt) (Thermo Fisher Scientific) or siCont (Ambion™ In Vivo Negative Control #1 siRNA (Thermo Fisher Scientific)) was mixed with transfection reagent (Invivofectamine 3.0 (Thermo Fisher Scientific)), and then the mixture was injected intravenously into the tails of mice (Day 0). One day after the injection, half of the tumor was cut off to facilitate metastasis (day 1), and metastatic conditions in the lungs and brain were evaluated after another 12 days (day 13). An additional experiment with the same timeline was performed to analyze the expression dynamics of S100 family genes in the lungs and brain; these organs were resected at 7 days after the intradermal inoculation of cells into the ear.

### 4.11. Statistical Analysis

Data are expressed as means ± SD. We used a simple pairwise comparison with Student’s *t*-test (two-tailed distribution with two-sample equal variance). Probability (*p*)-values < 0.05 were considered significant.

## Figures and Tables

**Figure 1 ijms-23-10300-f001:**
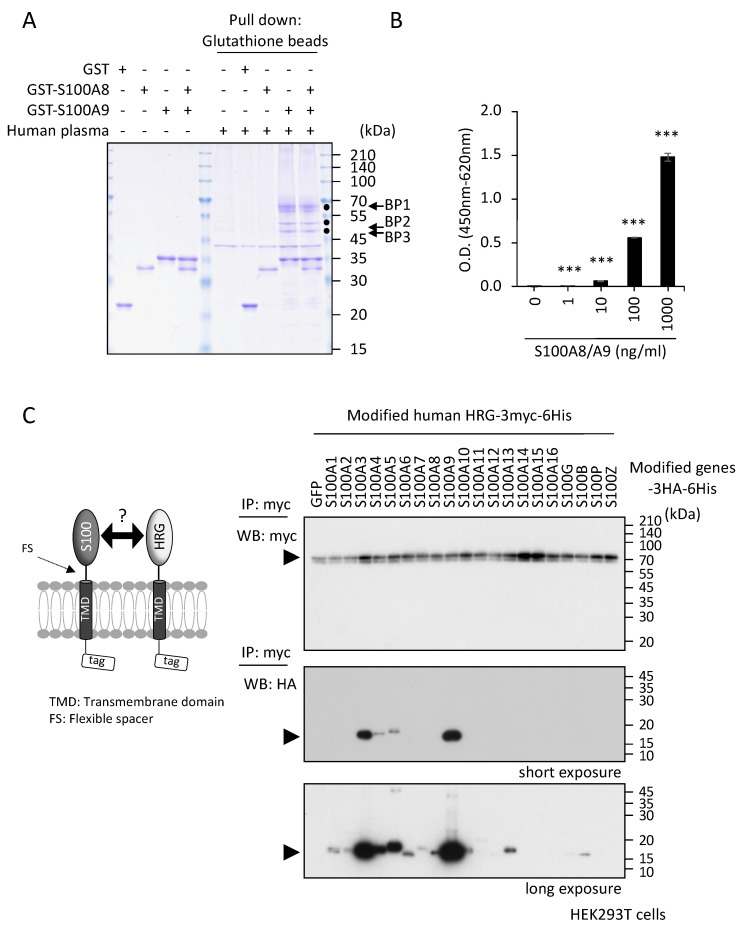
Identification of HRG as a binding protein to S100A8/A9. (**A**) Recombinant proteins (GST, GST-S100A8, GST-S100A9) were prepared from the *E. coli* expression system. Ten micrograms of each individual single protein or a preincubated mixture of GST-S100A8 and GST-S100A9 (GST-S100A8/A9) was incubated with human plasma (1 mL) collected from a healthy volunteer donor. After incubation for 30 min at room temperature under gentle shaking, the GST proteins were all pulled down by the addition of glutathione-conjugated Sepharose beads. After the beads were washed extensively, the bound proteins were eluted with glutathione and subjected to SDS-PAGE for analysis. The Coomassie brilliant blue (CBB)-stained gel displayed three clear bands (BP1, 2, and 3) that co-shed with GST-S100A9 and GST-S100A8/A9. (**B**) Interaction of S100A8/A9 with HRG was confirmed by ELISA. A 96-well plate coated with the highly purified human recombinant HRG protein (100 µg/mL) was incubated with the indicated concentrations of biotinylated human recombinant S100A8/A9 protein after blocking with a chemical-based reagent, Blockmaster DB1130, that is very good for quenching the nonspecific binding of S100A8/A9. The binding of S100A8/A9 to HRG was detected by treatment with HRP-conjugated streptavidin and the chemical reaction between the HRP and the substrate used. The background values of S100A8/A9 binding from the wells without HRG coating were deducted. Data are expressed as optical density (O.D.) means ± SD. *** *p* < 0.001 by Student’s *t*-test. (**C**) Schematic drawing of the method used to detect the interaction between S100 family proteins and HRG on the HEKT293 cell surface (left). To effectively increase the probability of interaction between S100 proteins and HRG in the extracellular space, all proteins were modified to express in a membrane-anchored form, so that their density was much greater on the narrow cell surface than in the vast extracellular space. HEK293T cells were co-transfected with myc-tagged modified HRG and HA-tagged S100 family-expressing vectors. After the immunoprecipitation of the expressed cell-surface HRG with myc antibody-conjugated beads, the interacting cell surface S100 proteins were detected by the HA antibody (right).

**Figure 2 ijms-23-10300-f002:**
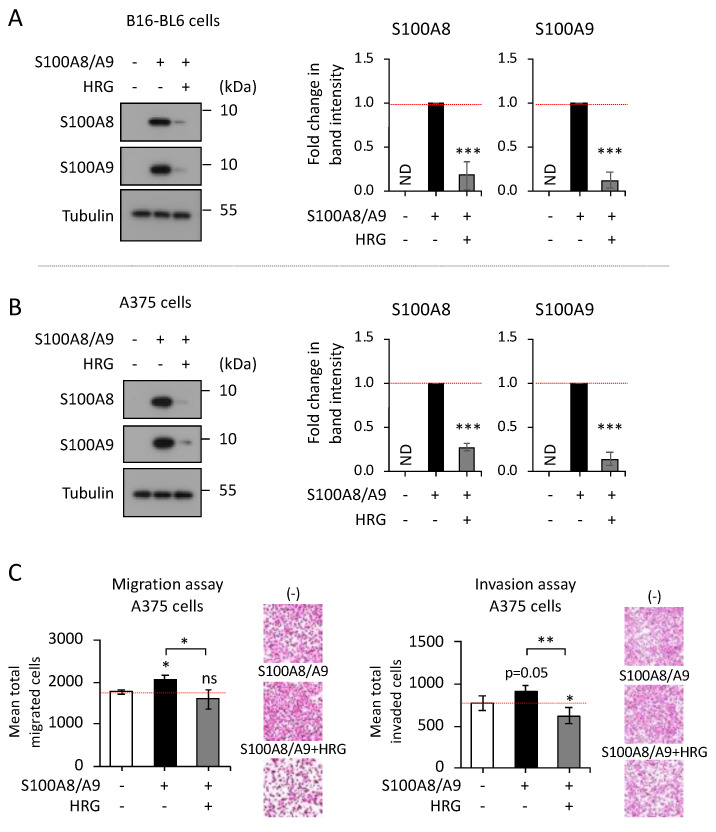
Prevention of the S100A8/A9-mediated migration and invasion of melanoma cells by extracellular HRG. (**A**,**B**), B16-BL6 cells (**A**) and A375 cells (**B**) were treated with S100A8/A9 (0.1 µg/mL) in the presence or absence of HRG (1 µg/mL) for 1 h. The treated cells were washed with PBS and collected as cell pellets. The lysed cell pellets were then subjected to SDS-PAGE followed by Western blotting for the detection of cell-bound ectopic S100A8/A9 protein. The Western blotting was repeated three times for the distinct samples prepared from the independent experiments, and the results were shown as the representative images ((**A**,**B**), left) and the quantified data ((**A**,**B**), right). For the quantification, the band intensities were all measured using ImageJ software (https://imagej.nih.gov/ij/, accessed on 9 August 2022). The intensities of the individual target bands (S100A8 and S100A9) were then calibrated to those of the corresponding tubulin bands (internal control) and were presented as the fold change compared with those of the indicated setting at the single treatment with S100A8/A9, whose values were set as 1.0. Data are means ± SD, *** *p* < 0.001 by Student’s *t*-test (*n* = 3), ND: not detected. (**C**) Experimental settings similar to those described in (**B**) were applied to evaluate HRG’s effects on S100A8/A9-mediated cancerous events in culture. Migration (**top**) and invasion (**bottom**) were evaluated using a Boyden chamber set with unburied and buried cell transmembrane with Matrigel, respectively. Cells were placed in the top chamber, and S100A8/A9 (1 µg/mL) with HRG (100 µg/mL) or BSA (a negative control, 100 µg/mL) was added to the bottom well. Eighteen hours later, migrating (**top**) and invaded (**bottom**) cells were detected with H&E staining (rightmost side, representative images) and then counted (left side, quantified data). Data are means ± SD. * *p* < 0.05 and ** *p* < 0.01 by Student’s *t*-test.

**Figure 3 ijms-23-10300-f003:**
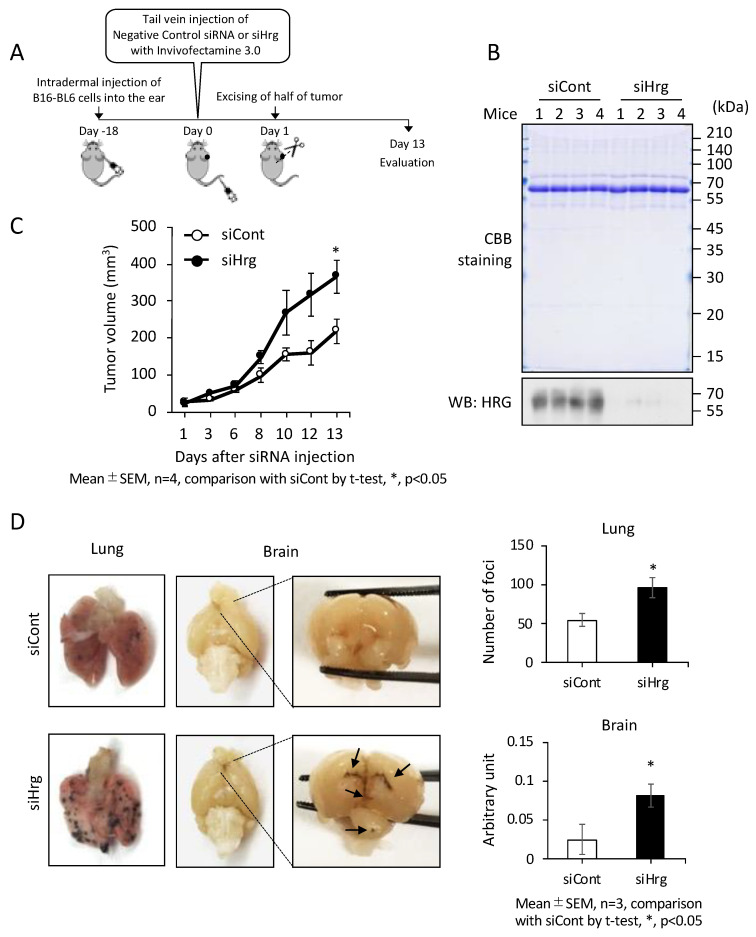
Evaluation of the anti-cancer role of the intrinsic HRG in a mouse model. (**A**) Schematic representation of the timeline of autonomous metastatic tumor progression in C57BL/6J mice burdened with mouse melanoma (B16-BL6) cells). siHrg or siCont was injected into mice 18 days after the inoculation of melanoma cells (see Materials and Methods for details). (**B**) On day 13, blood specimens (0.5 µL) were collected from the hearts of all mice and used to confirm that the HRG protein level was lower in the siHrg-treated mice than in the siCont-treated mice. (**C**) The volumes of B16-BL6 cell derived tumors grown in the individual ears were measured on the indicated days. (**D**) Lung and brain metastases were monitored by assessing the region of black tissue (melanoma) in images of dissected mouse lungs and brain (**left**). The clear black foci in the lungs that were greater than 1 mm in diameter were counted as metastatic foci (**upper-right** image). To evaluate brain metastasis, the intensity of the black area was measured using ImageJ software (lower-right image). Arrows indicate the sites of metastasis. Data are means ± SD. * *p* < 0.05 by Student’s *t*-test.

**Figure 4 ijms-23-10300-f004:**
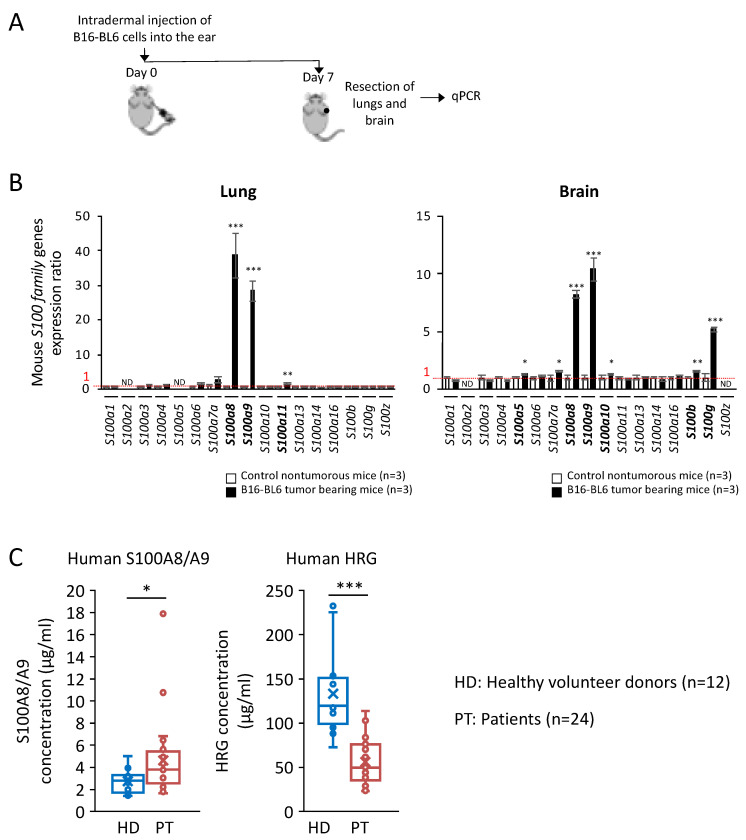
Profiling of the expression pattern of mouse S100 family genes in the premetastatic stage in the lungs and brain of C57BL/6J mice burdened with melanoma (B16-BL6 cells). (**A**) Lungs and brain were resected from C57BL/6J mice burdened with melanoma (B16-BL6 cells) on day 7 after inoculation of melanoma to the subcutaneous area of the ear. (**B**) Total RNAs prepared from the indicated organs (lungs and brain) were analyzed for the expression of S100 family genes by quantitative real-time PCR. Tbp mRNA was used as a control for the analysis. The data were expressed as an expression ratio in comparison with the respective values from the healthy lungs and brain of nontumorous control mice, which were set as 1. (**C**) S100A8/A9 and HRG levels in the plasma collected from brain-metastatic lung cancer patients. The backgrounds of patients are described in Materials and Methods. S100A8/A9 and HRG levels were measured with ELISA. Data in panels (**B**,**C**) are means ± SD. * *p* < 0.05, ** *p* < 0.01, and *** *p* < 0.001 by Student’s *t*-test.

**Table 1 ijms-23-10300-t001:** Disease characteristics.

Plasma		Healthy Volunteer Donors	Patients
n		12	24
Sex	Male/Female	9/3	13/11
Age, years	Median (range)	35 (24–72)	67 (46–78)
Histology of lung cancer	Ad/Sq/NEC	0/0/0	19/2/3
Stage of lung cancer	1A/1B/2A/2B/3A/3B/4A/4B		1/0/0/0/2/4/10/7

Ad: adenocarcinoma; Sq: squamous cell carcinoma; NEC: neuroendocrine carcinoma.

**Table 2 ijms-23-10300-t002:** RT-qPCR Primer Sequences.

Target	Forward (5′ to 3′)	Reverse (5′ to 3′)
Tbp	gggagaatcatggaccagaa	gatgggaattccaggagtca
S100a1	ctttctggcttcctggatgt	accagcacaacatactccttga
S100a2	gggagataaaggagcttttgc	tcttcaccttctcatcatctacgtt
S100a3	ccagtcggagctcaagga	ttgtagtcacactcccggaac
S100a4	tcagcacttcctctctcttgg	tttgtggaaggtggacacaa
S100a5	ggagttgatcaagacagagctga	tccaggctcttcatcaagttatc
S100a6	aggaaggtgacaagcacacc	agccttgcaatttcagcatc
S100a7a	tgcaccaagagcaacagact	gtctggcatgactgatggac
S100a8	tccttgcgatggtgataaaa	ggccagaagctctgctactc
S100a9	gacaccctgacaccctgag	tgagggcttcatttctcttctc
S100a10	gttccctgggtttttggaa	aagcccactttgccatctc
S100a11	gggaaggatggaaacaacact	caccaggatccttctggttc
S100a13	ccttgcctggtgcttataaactt	tgatgtccacacagattgacc
S100a14	atgggacagtgtcggtcag	gtgtctcaatggccctctct
S100a16	gatcagcaagtccagcttcc	ccaggttctggatgagcttg
S100b	aacaacgagctctctcacttcc	cgtctccatcactttgtcca
S100g	gctctccaaggaggagctaaa	cagctccttaaagagattgtcca
S100z	actcacagagttcctcacatgc	tccacttcgttgtctttattgg
Hrg	caccaactgtgatgcttctga	agtagtagactgtggccgttcc

## Data Availability

The original contributions presented in the study are included in the article; further inquiries can be directed to the corresponding author.

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
