# Peer review of "Histidine-Rich Glycoprotein Suppresses the S100A8/A9-Mediated Organotropic Metastasis of Melanoma Cells"

_ijms, 2022, doi:10.3390/ijms231810300_

Round 1

Reviewer 1 Report

Thank you for allowing me to review this original article entitled "Histidine-rich glycoprotein suppresses the S100A8/A9-mediated organ-tropic metastasis of melanoma cells." The topic is intriguing, but as the authors correctly said, molecular and cellular aspects of metastatic processes are not entirely known. Nevertheless, the present study is well-conducted, and the results support the authors' hypotheses in both in vitro and in vivo experiments. 

Authors find a plasma glycoprotein, histidine-rich glycoprotein (HRG),  inhibits the activity of extracellular heterodimer S100A8/A9. In vitro, adding HRG seems to decrease the S100A8/A9-mediated migration and invasion of A375 cells.   In the murine model, the plasma reduction of HRG is associated with an accelerated outgrowth of melanoma, and in siHrg-treated mice, it significantly induces lung and brain metastasis.  

The introduction section is well written, although I suggest underlining that findings related to S100A8/A9 are mainly in a preclinical phase. 

In the result section, the authors mention they  investigate the plasma

levels of S100A8/A9 and HRG in clinical patients who suffered from a brain metastasis of lung cancer, finding that S100A8/A9 was significantly elevated. Conversely, HRG was significantly reduced in patients with brain metastasis compared to healthy donors. It would be interesting to know if the same parameters are significant compared to lung cancer patients with no metastatic disease. Are S100A8/A9 and HRG measured in melanoma patients?  

In the discussion and conclusion section, the work tries to shed light on the shade of the metastatic process mediated by S100A8/A9. However, the authors clearly state study limitations. Indeed, the induction of S100A8/A9 in the premetastatic phase in the brain and the lung is exciting and deserves more studies.

Author Response

Responses to Reviewer 1

General Comments:

Thank you for allowing me to review this original article entitled "Histidine-rich glycoprotein suppresses the S100A8/A9-mediated organ-tropic metastasis of melanoma cells." The topic is intriguing, but as the authors correctly said, molecular and cellular aspects of metastatic processes are not entirely known. Nevertheless, the present study is well-conducted, and the results support the authors' hypotheses in both in vitro and in vivo experiments. 

Authors find a plasma glycoprotein, histidine-rich glycoprotein (HRG), inhibits the activity of extracellular heterodimer S100A8/A9. In vitro, adding HRG seems to decrease the S100A8/A9-mediated migration and invasion of A375 cells.  In the murine model, the plasma reduction of HRG is associated with an accelerated outgrowth of melanoma, and in siHrg-treated mice, it significantly induces lung and brain metastasis.  

The introduction section is well written, although I suggest underlining that findings related to S100A8/A9 are mainly in a preclinical phase. 

Response: Thank you very much! We heartily appreciate your warm comment.

In the result section, the authors mention they investigate the plasma levels of S100A8/A9 and HRG in clinical patients who suffered from a brain metastasis of lung cancer, finding that S100A8/A9 was significantly elevated. Conversely, HRG was significantly reduced in patients with brain metastasis compared to healthy donors. It would be interesting to know if the same parameters are significant compared to lung cancer patients with no metastatic disease. Are S100A8/A9 and HRG measured in melanoma patients? 

Response: Thank you very much for your helpful comment. The specimens were from patients with only brain metastasized lung cancers, which we collected little by little. As the reviewer suggested, we need to obtain measurements in specimens from patients with nonmetastasized lung cancers. Unfortunately, at this time we don’t have any specimens of the kind. However, in future work, after we collect enough specimens for a measurement study, we will definitely try to obtain such measurements as the reviewer suggested. We sincerely hope the reviewer permits our present situation.

    For specimens derived from melanoma patients, there were only a few stocks, that is, two specimens in our bio-bank, so we were not able to show the measured result as a reliable readout in the formal manuscript even in the Supplementary Figure. However, with regard to the reviewer’s kind remark, we would like to describe the result in this letter. As shown in the below Figure, we found that S100A8/A9 and HRG levels are in an inverse relationship, as with the data in the cases of metastasized lung cancer patients. That is, when S100A8/A9 levels increase, HRG levels decrease.

In the discussion and conclusion section, the work tries to shed light on the shade of the metastatic process mediated by S100A8/A9. However, the authors clearly state study limitations. Indeed, the induction of S100A8/A9 in the premetastatic phase in the brain and the lung is exciting and deserves more studies.

Response: Thank you so much! We appreciate the wonderful comment.

Reviewer 2 Report

In the present study, the authors addressed the question of why melanoma cells are not metastasized into the brain in mice, despite the marked induction of S100A8/A9 in the brain. They identified HRG as the major target of S100A8/A9.

And they found that HRG prevents the binding of S100A8/A9 to the B16-BL6 melanoma cell surface via the complex formation. HRG also inhibited the S100A8/A9-induced migration and invasion of A375 melanoma cells. Knocked down HRG in mice bearing skin melanoma, metastasis to the brain and lungs was significantly enhanced. Furthermore, clinical examination of plasma S100A8/A9 and HRG levels showed that lung cancer patients with brain metastasis had higher S100A8/A9 and lower HRG levels than non-metastatic patients.

 1.   The manuscript requires close copy editing to correct frequent noun/verb agreement and grammar errors.

 2.   In figure2, it is recommended that the author do experiments with three replicates and do statistical analysis in vitro study.

 3.   In figure 4B, the authors detected the expression pattern of mouse S100 family genes in the premeta static stage in the lungs and brain of melanoma (B16-BL6 cells)-burdened C57BL/6J mice. It is better to provide the control group data.

Author Response

Responses to Reviewer 2

General Comments:

In the present study, the authors addressed the question of why melanoma cells are not metastasized into the brain in mice, despite the marked induction of S100A8/A9 in the brain. They identified HRG as the major target of S100A8/A9.

And they found that HRG prevents the binding of S100A8/A9 to the B16-BL6 melanoma cell surface via the complex formation. HRG also inhibited the S100A8/A9-induced migration and invasion of A375 melanoma cells. Knocked down HRG in mice bearing skin melanoma, metastasis to the brain and lungs was significantly enhanced. Furthermore, clinical examination of plasma S100A8/A9 and HRG levels showed that lung cancer patients with brain metastasis had higher S100A8/A9 and lower HRG levels than non-metastatic patients.

  1. The manuscript requires close copy editing to correct frequent noun/verb agreement and grammar errors.

Response: We are so sorry for the English problems throughout the text. The revised text was corrected by a reliable proofreading service (KN international Inc (https://www.kninter.co.jp)), which uses native English-speaking editors.

  1. In figure 2, it is recommended that the author do experiments with three replicates and do statistical analysis in vitro study.

Response: Thank you so much for the instructive suggestion, which we heartily appreciate and agree with. We worked into Figure 2 the quantified data with statistical analysis from results replicated three times.

  1. In figure 4B, the authors detected the expression pattern of mouse S100 family genes in the premetastatic stage in the lungs and brain of melanoma (B16-BL6 cells)-burdened C57BL/6J mice. It is better to provide the control group data.

Response: Thank you so much! We agree with your suggestion. The data were shown by arbitrary units of fold induction to each control mean, which was set as 1. The real means of the control were obtained by the same qPCR using lungs or brains resected from nontumor-bearing mice. To clarify the point, we provided the control group data as the reviewer suggested. Thank you very much again.

Round 2

Reviewer 2 Report

The authors answered all my questions.